# Using Moderate Transgene Expression to Improve the Genetic Sexing System of the Australian Sheep Blow Fly *Lucilia cuprina*

**DOI:** 10.3390/insects11110797

**Published:** 2020-11-13

**Authors:** Ying Yan, Megan E. Williamson, Maxwell J. Scott

**Affiliations:** 1Department of Insect Biotechnology in Plant Protection, Institute for Insect Biotechnology, Justus-Liebig-University Giessen, Winchesterstraße 2, 35394 Giessen, Germany; 2Department of Entomology and Plant Pathology, North Carolina State University, Campus Box 7613, Raleigh, NC 27695-7613, USA; mewilli9@ncsu.edu

**Keywords:** livestock pest, *Lucilia cuprina*, genetic control, pro-apoptotic gene, sterile insect technique, genetic sexing, tetracycline-off system, insect transgenesis

## Abstract

**Simple Summary:**

Populations of pest insects can be suppressed through repeated mass releases of sterilized insects. This is particularly effective if only sterile males are released. We previously developed several genetically modified strains of the Australian sheep blowfly that produce only males when raised on diet that lacked tetracycline. A disadvantage of the some of the engineered strains was that females would lay few eggs unless fed a diet with a low dose of tetracycline. In this study we show that effective male-only strains can be made by combining driver/effector lines that have moderate transgene expression/activity. Furthermore, the strain does not require tetracycline in the adult diet for female fertility. This “moderate expression/activity” strategy could be more generally applied to other pests that can be genetically modified.

**Abstract:**

The sterile insect technique (SIT) is a promising strategy to control the Australian sheep blow fly *Lucilia cuprina*, a major pest of sheep. We have previously developed a transgenic embryonic sexing system (TESS) for this pest to facilitate the potential SIT application. TESS carry two transgenes, a *tetracycline transactivator* (*tTA*) driver and a tTA-activated pro-apoptotic effector. TESS females die at the embryonic stage unless tetracycline is supplied in the diet. However, undesired female sterility was observed in some TESS strains without tetracycline due to expression of tTA in ovaries. Here we investigate if TESS that combine transgenes with relatively low/moderate expression/activity improves the fertility of TESS females. tTA driver lines were evaluated for *tTA* expression by quantitative real time PCR and/or by crossing with a tTA-activated RFPex effector line. Fertility and lethality tests showed that a TESS strain containing a driver line with moderate *tTA* expression and an effector line showing moderate pro-apoptotic activity could recover the fertility of parental females and eliminated all female offspring at the embryonic stage. Consequently, such a strain could be further evaluated for an SIT program for *L. cuprina*, and such a “moderate strategy” could be considered for the TESS development in other pest species.

## 1. Introduction

The Australian sheep blow fly *Lucilia cuprina* Wiedemann (Diptera: *Calliphoridae*) is a major livestock pest that poses a threat to the sheep industry in Australia, New Zealand and Africa [1,2,3]. *Lucilia cuprina* prefers warmer temperatures but could be expanding its range into historically cooler areas, presumably due to the changing climate [4]. *Lucilia cuprina* is a facultative parasite with females laying their eggs in an open wound or orifice and the developing larvae consuming the flesh of the host animal. Thus, in addition to causing considerable economic losses, *L. cuprina* is a major concern for animal welfare [5,6]. The sterile insect technique (SIT), which was successfully implemented to eradicate the New World screwworm *Cochliomyia hominivorax* Coquerel in North and Central America [7], was similarly proposed to battle against *L. cuprina* [8]. For the ongoing screwworm SIT program, insects are sterilized by radiation and both sexes are released into the field to mate with their wild counterparts. This approach is not optimal since sterile females and males that are released together can mate with each other, thus reducing the efficiency of suppression [9,10]. Indeed, large field cage tests demonstrated that male-only sterile releases of the Mediterranean fruit fly (*Ceratitis capitata* Wied.) increased the efficiency of population suppression by three–five-fold relative to bisexual releases [11].

To effectively remove females from the releasing population, a two-component genetic system was demonstrated in *Drosophila melanogaster* in which all females were eliminated at the pupal stage [12,13]. Subsequently, a transgenic embryonic sexing system (TESS) was developed for tephritid fruit flies including *C. capitata* and the Caribbean fruit fly *Anastrepha suspensa* [14,15]. We previously reported the generation of TESS strains for *L. cuprina*, which can eliminate all females at an early stage and lead to a male-only population [16,17]. TESS employs the tetracycline-off (Tet-off) system, which contains a “driver” component that expresses the *tetracycline transactivator* (*tTA*) gene under the control of a promoter mostly active in early stage embryos and a tTA-activated “effector” component that expresses a pro-apoptotic gene, *L. sericata hid* (*Lshid*). In the absence of tetracycline, tTA binds to the tet operator (tetO) sequences upstream of the *Lshid* exons and activates expression of *Lshid* in embryos at the cellular blastoderm stage. The *Lshid* effector gene contains the sex-specific first intron from the *C. hominivorax transformer* gene [18]. After RNA processing, only female transcripts encode functional LsHID protein due to the sex-specific RNA splicing. Consequently, all females die at either embryo [16] or early larval stage [17] depending on the promoter that controls the timing of *tTA* expression during early development.

Ideally, the early promoter for *tTA* regulation should only be active during embryogenesis and result in all females dying before the feeding stage. Since larval diet is a major cost in SIT programs, early female lethality would lead to significant savings in the rearing costs of the program. Typically, promoters from cellularization genes such as *bottleneck*, *nullo* and *serendipity alpha* are used to drive early embryo *tTA* expression [14,15]. However, we have found that the promoters from the *L. sericata bottleneck* (*Lsbnk*) and *spitting image* genes (*Lsspt*) genes were also active at later stages of development including adults. Consequently, females from the TESS using these promoters were sterile and short-lived, possibly due to the LsHID production in the ovaries that was triggered by the “leaky” tTA expression [16,17]. Although female viability and fertility were restored by adding a high concentration of tetracycline to the diet until egg laying, such treatment resulted in any eggs laid carrying sufficient tetracycline to inhibit tTA in the developing embryo. Thus, it was necessary to find a feeding regimen of a low dose of tetracycline for only a few days after eclosion, which was sufficient for female fertility but would not prevent the female embryo lethal system from engaging in the next generation.

The activities of both driver and effector transgenes were impacted by negative genomic position effects, as is commonly associated with the *piggyBac*-mediated transformation system [16,19]. Indeed, some combinations of driver and effector lines showed high female survival in TESS from different species [14,15,17,20]. We considered the possibility that an effective TESS could be made by combining an effector line with moderate activity with a driver line that had low expression in adult females due to position effects. Here we have characterized several driver lines by quantitative real time (qRT)-PCR as well as by crossing with a tTA-inducible tetO-RFPex reporter line [21]. Multiple TESS strains were assembled using different driver and effector lines [16,17]. Fertility and lethality tests were performed to assess the performance of these TESS strains. We show that by combining “moderate” driver and effector lines an effective TESS strain can be made that does not require tetracycline in the adult diet.

## 2. Materials and Methods

### 2.1. Fly Rearing and Germ-Line Transformation

The LA07 wild type strain of *L. cuprina* was maintained and transformed as previously described [16]. Specifically, embryos were microinjected with a mixture of synthesized *piggyBac* RNA helper (300 μg/mL), *Lchsp83-pBac* helper (200 μg/mL) and *pBac*[Effector-*rpr*] plasmid (700 μg/mL). First instar larvae showing transient expression of the DsRedex2 marker were selected and raised on raw 93% ground beef. G_0_ adults were crossed to wild type flies and offspring screened for expression of the fluorescent marker at late embryo/first instar stages. Homozygous individuals were selected at the wandering third instar larval stage based on fluorescence intensity and bred to create a stable line.

### 2.2. Plasmid Construction

An Lsrpr fragment including 5′UTR and 3′UTR [1] was amplified from L. sericata embryo cDNA using primers Ls-rpr-5UTR-F: 5′-GAGTTTCCATCTAGCAAACAAACAAT-3′ and Ls-rpr-5UTR-R: 5′-TTATTTTTTAGCGGGTTTCACTTTT-3′, and cloned into pGEM-T (Promega) to form pGEM-Lsrpr. The Chtra intron fragment was amplified from pBS-FL3 [2] using primers NSWtra-StuI: 5′-TTTTAGGCCTCTAATTTTTTGAGCAACATT-3′ and Lsrpr-NSWtra: 5′-CGGGTATATAGAATGCTACAGCCTAAACATAGAAAAGAATAATAAATTTATCATACA-3′, and an Lsrpr fragment was amplified from pGEM-Lsrpr using primers NWStra-Lsrpr: 5′TGTATGATAAATTTATTATTCTTTTCTATGTTTAGGCTGTAGCATTCTATATACCCG-3′ and LsrprTAA-BgIII: 5′-TCTGAGATCTTTATTTTTTAGCGGGTTTCACTTTTTTGTTG-3′. The two PCR products were purified and combined as template for a second round of PCR, from which a Chtra_intron-ATG-Lsrpr fragment was amplified using primers NWStra-StuI and LsrprTAA-BgIII. Then the amplified fragment was ligated to pBS-FL3 [3] using unique StuI and BgIII sites to generate pBS-tetO21-Dmhsp70-Chtra_intron-ATG-Lsrpr-SV40 (pBS-effector-rpr). To replace the Dmhsp70 core promoter with the Lchsp70 core promoter in the plasmid pBS-effector-rpr, an Lchsp70-promoter-Chtra_intron fragment was cut from pBS-FL11 [3] and inserted into pBS-effector-rpr using unique BamHI and StuI sites. Finally, the tetO21-Lchsp70-Chtra_intron-ATG-Lsrpr-SV40 cassette was excised by digestion with XhoI and NotI and cloned into the unique XhoI and PspOMI sites in the piggyBac transformation vector pB[Lchsp83-DsRedex2] [4].

### 2.3. RNA Isolation and qRT-PCR Analysis

Samples from different developmental stages and sexes were collected as before [16,17]. Total RNA was extracted using the RNeasy^®^ Mini Kit (QIAGEN) according to the manufacturer’s instructions. Isolated RNA was subsequently treated with the RNase-Free DNase Set (Qiagen). Then, 5 μg RNA was used to synthesize cDNA using Superscript III First Strand Synthesis Supermix (Invitrogen) following the manufacturer’s instructions. A control reaction lacking Superscript III was included for each preparation. qRT-PCR was performed as previously described with *LcGST1* serving as the reference gene [16,17].

The 2^-∆∆Ct^ method was used with all samples relative to the *tTA* expression in 2 days old females. The primers for the reference gene are *LcGST1*-F-GCCAGTGTCAGCACCTTTG and *LcGST1*-R-GCAACCTTCCCAGTTTTCATC. The primers for DR2 and DH1 strains are *tTAo*-F-TGTTGAATGAAGTGGGTATTGAAGGATTGACTACTCG and *tTAo*-R-CCAAAGGGCAAAAGTGGGTGTGATGTCTATC. The primers for DH4 strain are *tTAV*-F-TCTTGCGTAATAATGCCAAATCCTTCCG and *tTAV*-R- CCAACACACAGCCCAATGTAAAATGACC.

### 2.4. Confocal Imaging to Assess Ovary-Specific Expression of tTA

The confocal images were acquired as previously described [21]. Briefly, homozygous males from one of the driver lines were crossed with homozygous virgin females from a tetO-RFPex reporter line. Ovaries were dissected from female offspring at four and eight days following eclosion and fixed by submerging in 4% paraformaldehyde (Biotium). Images were captured on a Zeiss LSM 880 confocal microscope using a 40× lens and the Argon and 561 lasers to capture green and red fluorescence, respectively. Additionally, all developmental stages were screened for red fluorescence using a Leica M205 FA Fluorescent Microscope.

### 2.5. Female Lethality Test and Embryo-Specific Lethality Assessments

Double homozygous (DH) strains which are homozygous for both the driver and effector transgenes, were generated using diet supplemented with 100 μg/mL tetracycline as previously described [16]. Specifically, homozygous virgin females from the effector lines were crossed with homozygous males from driver lines to generate double heterozygous strains. The double heterozygous strains were inbred and their progeny screened to select only individuals homozygous for both the driver and effector by epifluorescence microscopy based on fluorescence intensity. For female lethality tests, 10 pairs of newly emerged adults from a DH strain were used for each rearing bottle. Flies in these bottles were supplied with either 100 μg/mL tetracycline water for 8 days (+W), tetracycline-free water (−W) for 8 days or tetracycline water at a low concentration for two days then switched to tetracycline-free water for 6 days (+ −W). Embryos were then collected from each bottle and larvae were reared on either tetracycline-free meat (–M) or meat containing 100 μg/g tetracycline (+M). For staged lethality tests, 1000 embryos from the first egg lays of each DH strain were collected and reared on the tetracycline feeding regimen indicated. The number of 1st instar larvae, 3rd instar larvae, pupae, and adult males and females were monitored.

### 2.6. Statistical Analysis

All statistics were carried out using SigmaPlot v14 (Systat Software). The differences in the *tTA* expression level or number of flies from multiple sources were tested by one-way analysis of variance (ANOVA) and means were separated using the Holm–Sidak method. The number of flies from two sources were analyzed using two-sample *t*-tests as were the mean values from the number of male and female offspring from female lethality tests.

## 3. Results

### 3.1. A Moderate Driver Line with Reduced Expression in Ovaries

Previously reported TESS were made by combining a *Lsbnk-tTA* driver (DR2) or *Lsspt-tTA* driver (DR3) line with a tTA-activated *Lshid* effector (EF1 or EF3) [16,17]. Effector one (EF1) lines had a mutant version of *Lshid*, *Lshid^Ala2^*, that was predicted to be resistant to inhibition by MAP kinase since the two conserved predicted MAPK phosphorylation sites (PPSP and PPTP) were mutated to encode Alanine rather than Serine or Threonine [22,23]. EF3 lines carried a wild type version of *Lshid*. Three DR2 lines (#6, #7, #8) were analyzed here for *tTA* expression in early embryos and young adult females using qRT-PCR. DR2#7 had the lowest expression levels and DR2#8 the highest (Figure 1a). Specifically, *tTA* expression in early embryos (2 h old) and young females (2 days old) of DR2#7 were 2.1 (*p* = 0.440, One-way ANOVA) and 2.0 (*p* = 0.837, One-way ANOVA) times lower than these in DR2#6, and 2.7 (*p* = 0.067, One-way ANOVA) and 2.6 (*p* = 0.623, One-way ANOVA) times lower than in DR2#8, respectively (Figure 1a). In addition, *tTA* expression levels at different developmental stages were measured in two previously assembled TESS strains [16,17] (Figure 1b). In these strains, DR2#6 was combined with the effector EF3E and DR3#4 with the EF1-12 effector line. Both TESS were effective producing only males when raised on diet that lacked tetracycline. However, females produced very few eggs if tetracycline was not included in the adult diet [16,17]. The DR3#4;EF1-12 strain showed higher *tTA* levels than in the DR2#6;EF3E strain at all stages examined (Figure 1b).

We recently verified that the DR2#6 and DR3#2 driver lines express tTA in ovaries through crossing with a tetO-RFPex reporter line [21]. Thus, the undesired sterility of females from the previous TESS that used these drivers was likely due to the activation of the strong pro-apoptotic *Lshid* gene in the ovaries (Figure 2a). We hypothesized that an effective TESS could be made by combining a driver line that had low expression in adult females with an effector line that had low basal activity due to negative position effects (Figure 2a). To evaluate such a “moderate strategy”, DR2#7, which showed the lowest *tTA* expression in adult females (Figure 1a), was crossed with the tetO-RFPex line to evaluate if DR2#7 would trigger the expression of the red fluorescent protein gene in ovaries. Red fluorescence was readily observed in the germarium of 4- and 8-day-old female offspring of a cross of the DR3#4 driver with the tetO-RFPex line (Figure 2b, panels b1 and b2). Very weak fluorescence was observed in the germarium region of ovaries from 4-day-old females, and no red fluorescence was observed in ovaries from 8-day-old female offspring of a cross of the DR2#7 driver with the tetO-RFPex line (Figure 2b, panels b3 and b4). Red fluorescence was observed in the embryos from a cross with either driver and tetO-RFPex (data not shown), confirming *tTA* expression at the early stage. In addition, red fluorescence was observed in the 1^st^ and 3^rd^ instar larvae from the DR3#4 cross, but not at these stages from the DR2#7 cross (data not shown). The red fluorescence observed in DR3#4;tetO-RFPex ovaries supports our earlier conclusion that DR3#4;EF1-12 females were likely sterile because of activation of the *Lshid* gene by tTA [17]. Meanwhile, the very low red fluorescence in the ovaries of DR2#7;tetO-RFPex females suggested that the DR2#7 line could be used to make an effective TESS as part of the “moderate” strategy (Figure 2a).

### 3.2. The Combination of Selected Driver and Effector Lines Determines Female Fertility and Lethality

Multiple strains were assembled by combining different driver and effector lines that had varying levels of expression or activity (Table 1). For effectors, we selected the lines EF3F, EF3A and EF1-13, which exhibited low, moderate and strong female-specific lethality, respectively, when previously crossed to DR2#6 [16]. For drivers, we selected the lines DR2#7 and DR2#8, as tTA is expressed almost three times higher in line #8 than #7 (Figure 1a). Strains were bred to homozygosity for both driver and effector transgenes. Lethality tests were performed without tetracycline in the larval diet and with or without tetracycline added to the diet of the parental generation. The results show that the DH strains with the weak EF3F effector were fully viable and fertile without any tetracycline in the adult diet; however, female survival was high with either DR2#7 or DR2#8 (Table 1). With the moderate EF3A effector, if tetracycline was omitted from the adult diet, DR2#7; EF3A females were fertile while DR2#8; EF3A were sterile. Addition of 1 or 3 μg/mL tetracycline to the adult diet for the first two days after eclosion slightly increased pupal and male production of DR2#7; EF3A. However, it was necessary to supply DR2#8; EF3A females with a diet containing 50 μg/mL tetracycline to completely rescue female fertility (Table 1). The combination of DR2#8 with EF1-13 required the adult diet be supplemented with 100 μg/mL tetracycline to restore pupal production. Nevertheless, any tetracycline passed on from DR2#7; EF3A or DR2#8; EF1-13 females was insufficient to rescue any female offspring that were reared on diet without tetracycline (Table 1). In addition, one tetO-*Lsrpr* effector line (EF4) was generated by *piggy*Bac-mediated transformation and then combined with DR2#6 or DR3#4 driver lines. Adult females from these strains were viable and fertile without any tetracycline supply; however, all or most of females survived in the next generation (Appendix A). Specifically, in the absence of tetracycline, the DR2#6; EF4 strain produced similar numbers of male and female offspring (*p* = 0.134, t = 1.877, d.f. = 4), while the DR3#4; EF4 strain produced significantly more males than females (*p* = 0.013, t = 4.298, d.f. = 4) (Appendix A).

### 3.3. Female Embryo-Specific Lethality of DH Strains with EF3A

The DR2#7; EF3A and DR2#8; EF3A DH strains were further evaluated under different tetracycline feeding schemes. Both strains produced similar numbers of males and females on diet with 100 μg/mL tetracycline (*p* = 0.189, One-way ANOVA; Figure 3, +W, +M). Most females died if the parental generation was fed a high dose of tetracycline (100 μg/mL) but the larval diet lacked tetracycline (Figure 3, +W, −M), suggesting that the maternal tetracycline could rescue some female offspring. For DR2#7; EF3A, a maternal supply of tetracycline was essential for female survival as none were rescued by adding tetracycline to the larval diet (Figure 3a −W, +M), indicating all females died before the feeding stage. Importantly, DR2#7; EF3A females were viable and fertile without any tetracycline supply, and produced only male offspring (Figure 3a −W, −M). However, DR2#7; EF3A was not as productive as DR2#7; EF3F or wild type, producing less than a third of the number of pupae from a similar number of egg clutches (Table 1). For DR2#8; EF3A, females produced few offspring with a low dose of tetracycline (3 μg/mL), while a high dose of tetracycline (50 μg/mL) was needed to restore female fertility while also producing only male offspring in the next generation (Figure 3b, + −W, +M). Staged lethality tests were carried out for strain DR2#7; EF3A to further assess when females die (Figure 4). Compared to the number of first instar larvae that emerged from 1000 embryos from the constant tetracycline condition (+W, +M), only about half the amount were found on diet without tetracycline (−W, −M) or tetracycline only for the larval diet (−W, +M). This result suggests females died at the embryo stage.

## 4. Discussion

The previously developed *L. cuprina* TESS strains with DR2 or DR3 drivers worked efficiently to eliminate females (100% dominate lethality) at the embryo stage [16] or early larval stages [17]. However, the requirement for supplemental tetracycline to the adult diets of these TESS strains would add extra cost in labor and material for an SIT program, that involves mass-rearing of millions of insects. In addition, the maternally supplied tetracycline may partially suppress the lethal system in the release generation and lead to female survivors, resulting in a reduction in the program’s control efficiency [24,25]. Moreover, reducing the usage of tetracycline could be beneficial since antibiotics eliminate some bacterial populations in insects with a consequent negative impact on fitness and male performance [26,27,28]. Out of the seven DH strains made in this study (Table 1, Appendix A), we successfully identified one strain DR2#7; EF3A, which does not need an additional tetracycline supply for egg production but produces only male offspring. Therefore, this is an improvement over the previously reported TESS strains that used the DR2 driver as well but had undesired sterility. The male production of DR2#7; EF3A is relatively low compared to some other TESS strains (Table 1), but could be compensated by seeding more eggs during mass rearing since females can be eliminated before the feeding stage.

The *Lsrpr* gene was less effective at inducing widespread cell death compared to *Lshid* when previously tested in *Drosophila melanogaster* using the GAL4-UAS system [22]. The *A. suspensa rpr gene* (*Asrpr*) also showed weaker proapoptotic activity than that of *Ashid* when tested in *D. melanogaster* S2 cells or the *A. suspensa* cell line UFENY-AsE01 [29]. Here we found an *Lsrpr* line in *L. cuprina* had no or very low female lethality when combined with DR2#6 or DR3#4 driver lines (Appendix A). In contrast, most of the DR2/DR3 and EF1/EF3 combinations tested previously were females lethal in the double heterozygous condition [16,17]. Thus, the results are consistent with our earlier observations that pro-apoptotic activity of *Lsrpr* was less than *Lshid.* However, it is possible that the low activity of the *Lsrpr* line is due to a negative position effect and thus additional transgenic lines would be required to confirm the low pro-apoptotic activity of *Lsrpr in L. cuprina.* While *Lsrpr* was relatively ineffective, some other strategies could be considered to build a moderate effector. For example, the EF3 construct comprised 21 copies of the tetO and the *hsp70* core promoter from *L. cuprina* (tetO_21_- *Lchsp70-Lshid*) [16], and previous studies showed that the seven copies of the tetO could also be used for embryonic lethality [14,15,30] and the *Dmhsp70* core promoter mediated weaker effector gene activity compared to *Lchsp70* core promoter in *L. cuprina* transgenic sexing strains [31]. Thus, by reducing the number of tetO copies in the effector and using a less active core promoter the expression of *Lshid* could be less responsive to tTA. Furthermore, since multiple copies of pro-apoptotic genes cause more intensive apoptosis than a single gene [29], it is possible to build a moderate effector with several copies of gene(s) showing weak pro-apoptotic activity that are co-expressed using the picornaviral self-cleaving 2A peptides [32]. Lastly, a moderate driver or effector can be generated by site-specific integration to a genomic site with known position effects on the transgene expression using either recombinase-mediated cassette exchange [33] or CRISPR/Cas9-mediated gene knock-in [34,35]. For example, a driver or effector cassette with strong expression/activity can be inserted into a genomic site with known negative position effects.

The lethality tests from different DH strains made in this study (Table 1) provide some practical information for the future development of functional TESS. First, an effector line with apparent low pro-apoptotic activity (EF3F or EF4) does not result in sufficient female lethality even when combined with a strong driver line (DR2#8). Second, TESS females are sensitive to the level of tTA expression since females from DR2#7; EF3A were fertile but DR2#8; EF3A females were completely sterile if the adult diet lacked tetracycline. Third, the results from the DR2#8; EF1-13 strain showed that an effective TESS can be made by combining a driver with high levels of tTA expression with a strong effector line but only if the adult diet is supplemented with a high dose of tetracycline for the first two days after eclosion. Fourth, crosses with a tetO-fluorescent protein reporter line can be used to identify tTA driver lines that have high expression in embryos but no expression in ovaries. Indeed, we recently found that driver lines made with the early promoters from the *L. cuprina nullo* or *Cochliomyia macellaria* CG14427 genes show high activity in embryos but no detectable *tTA* expression in female ovaries [21]. TESS assembled with either of these two drivers had high female fertility on adult diet without tetracycline and 100% female embryo lethality [21]. However, the identification and evaluation of an ideal promoter or pro-apoptotic gene for TESS development are largely dependent on the availability of genome resources and an efficient in vivo expression system, both of which could be very limited for some insect species. The *Lsbnk* promoter used in this study to drive tTA expression was obtained from *L. sericata* by a PCR-based genome walking approach [22]. With the availability of whole genome sequences for *L. cuprina* [36] and *C. hominivorax* [37], future driver strains can be made with gene promoters isolated from the species of interest.

## 5. Conclusions

Here we generated and evaluated several TESS strains for a potential genetic control program for *L. cuprina*. We showed one TESS strain, DR2#7; EF3A, that was an improvement over previous TESS strains that had undesired female sterility. This strain does not need supplemental tetracycline for egg production and eliminates all female offspring at the embryo stage. Therefore, DR2#7; EF3A is representative of the type of functional TESS strain that could lead to considerable savings in the running costs of an SIT program due to early and efficient female lethality and the generation of a male-only population for release.

## Data Availability

The NCBI accession number for EF4 is MW150993. The DR2, EF1 and EF3 plasmids were previously published [16] with accession numbers KT749916, KT749917 and KT749918 respectively.

## Figures and Tables

**Figure 1 insects-11-00797-f001:**
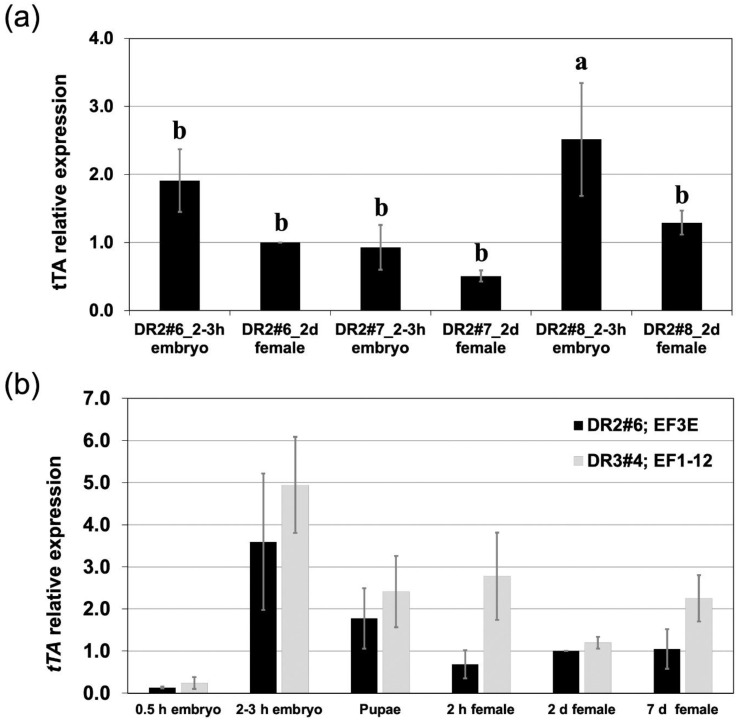
Evaluation of *tTA* expression from different driver lines by qRT-PCR. (**a**). Relative *tTA* expression levels in early embryo (2 h or h after egg laying) and young females (2 days or d after eclosion) of three DR2 driver lines were determined. The numbers with different lower-case letters are significantly different (*p* < 0.05, One-way ANOVA). (**b**) Relative *tTA* expression levels at different development stages of DH strains DR2#6; EF3E and DR3#4; EF1-12 are shown. The embryo samples were collected within 0.5 h or 2-3 h after egg laying and the female samples were collected 2 h, 2 d or 7 d after eclosion. For all qRT-PCR analysis, RNA levels are relative to the *LcGST1* reference gene and data are from three biological replicates.

**Figure 2 insects-11-00797-f002:**
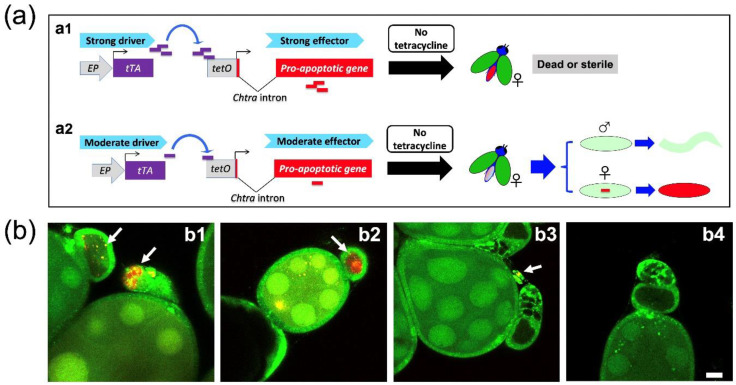
“Moderate driver and effector” strategy to improve the transgenic embryonic sexing system (TESS) of *L. cuprina*. (**a**) Schematic illustration for the transgenic embryonic sexing system (TESS) containing driver and effector components. The driver used an early promoter (EP) to drive the *tetracycline transactivator* (*tTA*), and the effector contains a sex-specific intron (*Chtra* which immediately follows the ATG translation start codon of pro-apoptotic gene) splicing cassette within a pro-apoptotic gene to achieve highly efficient female lethality early in the development of *L. cuprina*. Unless adults are fed diet with tetracycline, females are sterile and short-lived if the early promoter is also active in female ovaries and the effector is strongly activated by tTA (a1) [16,17,21]. However, a combination of moderate driver and effector could reduce such “leaky” cell death in females thus an additional tetracycline supply may not be necessary (a2). (**b**) Confocal images of fluorescent ovaries. The images were taken using dissected ovaries from 4 d (b1, b3) and 8 d (b2, b4) old females, which were generated by crossing homozygous males from DR3#4 (b1, b2) or DR2#7 (b3, b4) with homozygous virgin females from EF-RFPex [21]. DsRed expression (indicated by white arrows) was observed in the early portion of the developing eggs (germarium). The nurse cells showed green fluorescence from the marker. Scale bar: 20 μM.

**Figure 3 insects-11-00797-f003:**
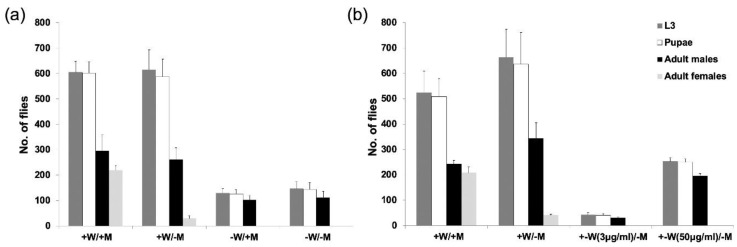
Female-specific lethality of TESS. The DR2#7; EF3A (**a**) and DR2#8; EF3A (**b**) strains were raised under different tetracycline feeding regimens. Containers were set with 10 pairs of adults and the number of third instar larvae (L3), pupae and adult male and female offspring counted. +W: water with 100 μg/mL tetracycline from day 1 (D1) to D8; + −W: water with limited tetracycline at indicated concentration from D1 to D2, then switched to water without tetracycline from D3 to D8; +M: ground meat (larval diet) with 100 μg/g tetracycline; −M: meat without tetracycline. Mean and standard error are shown from three replicate experiments.

**Figure 4 insects-11-00797-f004:**
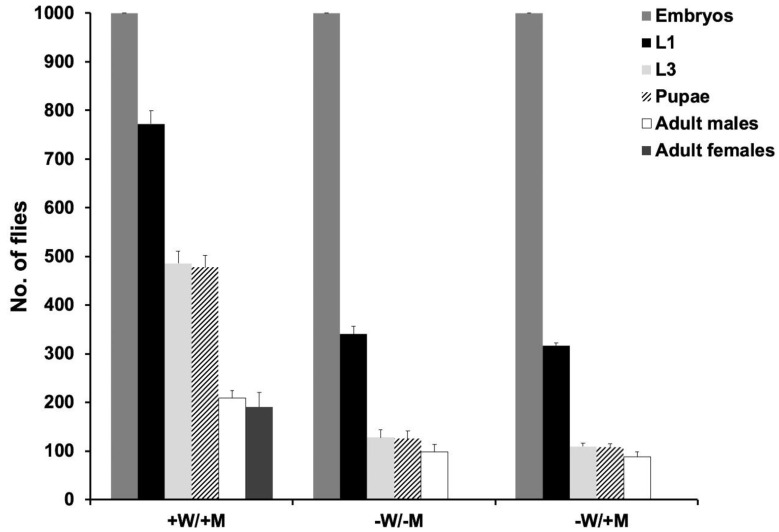
Staged lethality of the DR2#7; EF3A strain. For each test, 1000 embryos were collected and the numbers of first instar larvae (L1), third instar larvae (L3), pupae, adult males and adult females were recorded. Error bars show the mean and standard error (*n* = 3). +W/+M: parental water contained 100 μg/mL tetracycline and the larval diet (93% ground meat) contained 100 μg/g tetracycline; −W/−M: neither adult nor larval diets contained tetracycline; −W/+M: the adult diet lacked tetracycline but the larval diet contained 100 μg/g tetracycline.

**Table 1 insects-11-00797-t001:** Female fertility and lethality tests for different double homozygous strains.

Strains	Tet Water ^a^	Egg Clutches ^b^	#Pupae ^c^	#Male	#Female	Female (%)
Wildtype	0	4–8	843 ± 105	402 ± 46	376 ± 33	48.3
DR2#7; EF3F	0	4–8	503 ± 46	248 ± 28	160 ± 18	39.2
DR2#7; EF3A	0	4–8	143 ± 28	112 ± 25	0	0
DR2#7; EF3A	1	4–8	193	159	0	0
DR2#7; EF3A	3	4–8	197	153	0	0
DR2#8; EF3F	0	4–8	359 ± 38	176 ± 42	101 ± 11	36.3
DR2#8; EF3A	0	No eggs				
DR2#8; EF3A	1	1	5	3	0	0
DR2#8; EF3A	3	1–3	41 ± 7	31 ± 2	0	0
DR2#8; EF3A	50	4–8	249 ± 14	197 ± 9	0	0
DR2#8; EF1-13	1	No eggs				
DR2#8; EF1-13	3	No eggs				
DR2#8; EF1-13	10	No eggs				
DR2#8; EF1-13	20	No eggs				
DR2#8; EF1-13	100	1–3	321	296	0	0

^a^ Tetracycline at the indicated concentration (μg/mL) was supplied in the first two days after adult eclosion. ^b^ Tetracycline-free raw ground beef was used to collect eggs during a 24 h period on the 8th day after eclosion and the number of egg clutches were counted. ^c^ Data shown as mean ± standard deviation for three biological replicates; Otherwise the data was from one biological replicate.

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
