# Peer review of "Using Moderate Transgene Expression to Improve the Genetic Sexing System of the Australian Sheep Blow Fly Lucilia cuprina"

_insects, 2020, doi:10.3390/insects11110797_

Round 1

Reviewer 1 Report

The authors have addressed my initial review satisfactorily. Although I didn't suggest a title change, I do prefer the new title.

Author Response

no changes required

Reviewer 2 Report

Very interesting paper, providing an important improvement in the bio-technologies connected with SIT, for the control of this very important livestock pest. 

I made just few editing changes, addressed to follow the "taxonomic code": author's name; order & family).

Author Response

We have made the minor changes recommended by the referee.

Reviewer 3 Report

Authors have adequately addressed comments. I can now recommend publication of the manuscript. 

Author Response

No changes required

This manuscript is a resubmission of an earlier submission. The following is a list of the peer review reports and author responses from that submission.

Round 1

Reviewer 1 Report

The Australian blowfly, Lucilia cuprina, is a livestock pest that may potentially be controlled using SIT. This control strategy requires releasing sterilized males, so females need to be removed from mass rearing production. The authors have previously developed a transgenic embryonic sexing system (TESS) with “a tetracycline transactivator (tTA) driver and a tTA-activated pro-apoptotic effector”, however, leaky expression in the presence of tetracycline causes fertility problems among females (Yan 2015, 2017). This is undesirable for maintaining stocks. Here, the authors continue their efforts to characterise driver lines and build upon a recent similar publication (Yan et al. 2020).

The authors characterise strain DR2#7;EF3A, which has “moderate” expression of tTA and only produces male progeny in the absence of tetracycline (or low doses of tetracycline in adult water). Death in the female progeny occurs as embryos. Providing tetracycline in the fly water (100ug/ml) and larval diet (100ug/g) is able to rescue female survival.

This work contributes to the SIT literature in a positive way and represents the latest “chapter”. Readers will need a clear understanding of Yan et al’s previous work to fully appreciate this manuscript. To make the nomenclature/constructs more informative in this stand alone manuscript, I recommend including a schematic of these constructs, similar to Fig 2A in Yan et al. 2020, and potentially including a supplementary table that summarizes strain nomenclature, phenotypes, tTA expression and promoters.

Comments

Abstract

“The sterile insect technique (SIT) was proposed to control…” I’m not sure if this journal allows citations in the abstract but this would be an appropriate statement to cite.

“However, undesired female sterility was observed in some TESS strains due to expression of the tTA driver in ovaries”. Was this in the presence of tetracycline? If so, please indicate this.

“Here we investigate if TESS that combine transgenes with relatively low/moderate expression/activity improves TESS performance.” This could be made clearer by explicitly stating what the improvements should be.

L35, L37 – Use the genus when beginning a sentence where possible ( not L. )

L56: Consider replacing this line with something like “..depending on the promoter that controls the timing of tTA expression during early development”.

L115: Explain what the lines are “double homozygous” for (i.e. homozygous for both the driver and effector constructs). Although the authors state this was previously described, I think it would be valuable to briefly re-state how this is achieved.

Results

L137: A brief explanation of why LshidAla2 was predicted to be resistant to MAPK is important and explanation in the text would be helpful.

144: “two previously assembled TESS strains” – please provide a citation for these two strains and indicate whether female sterility occurs in the absence of tetracycline. It would also be helpful to explain the EFE3 and EF1-12 lines in the methods.

Table 1. It’s a slight concern that the DR2#8;EF3A experiments with tetracycline (1, 3 ug/ml) didn’t have biological replicates, but the data is consistent with zero tetracycline.

Figure 1. Different scales are used to compare tTA relative expression in Fig 1a and Fig 1b, which makes it difficult to compare the 2-3 h embryo data. Consider a linear scale for Fig 1b or providing a dashed line showing relative expression of 2.0 on the log scale.

L157: Consider deleting “On the other hand”

190: a new effector strain is described – EF4. This should be added to the methods along with EF1 and EF3. Although the results in Fig S1 are quite interesting, it breaks up the text and experiments regarding DR2#7; EF3A and DR2#8; EF3A. Perhaps this section could be moved elsewhere in the manuscript, explaining the key point that both the expression of the driver and effector are important for female lethality in the absence of tetracycline.

Figure 3. Controls have not been performed in this female specific lethality experiment. Are there numbers from a control strain that could be included for reference?

Constructs should be provided with accession numbers.

Reviewer 2 Report

i found the manuscript "Using insect strains with moderate driver and effectors trans gene activities to improve the genetic sexing system of the Australian she blow fly Lucilia cuprina" very interesting and very clear. The idea to use a transgenic embryonic sexing system is not innovative, but the way how the authors screened different TESS fly strains was very appropriate and effective.

The only comment I made was in the introduction: authors "jumped" from the "historical approach of SIT to their research, without any other examples of the use of transgenic embryonic sexing system on other diptera (see my comments referred to the lines 43 to 47).

Reviewer 3 Report

The manuscript by Yan et al. describes the improvement of the genetic sexing system of Lucilia cuprina that was generated by the same group earlier. A previous transgenic embryonic sexing system (TESS) in L. cuprina had resulted in undesired female sterility due to expression of the tTA driver in the ovaries. In the current study, the authors combine transgenes with relatively moderate expression, which results in improvement of the TESS performance. Data are solid, science is straightforward and good, and results may be useful in other groups’ efforts to solve similar problems.

I would be happy to recommend publication in Insects provided that among the reasons that may have caused the undesired female sterility the authors discuss the fact that all driver and effector constructs used promoters and genes from a sibling, yet distinct, species, L. sericata. In the last paragraph of the Discussion, the authors do mention that “the identification and evaluation of an ideal promoter or pro-apoptotic gene for TESS development are largely dependent on the availability of genome resources…”. Since the genome of L. cuprina is available, it is curious why the authors did not consider the possibility that some of the unexpected observations may be due to the use of non-species-specific constructs. In fact, since L. sericata promoters (Lsbnk and Lsspt) are leaky, the authors should argue why they have not considered using a species-specific (i.e. L. cuprina) early embryonic promoter to drive tTA expression.

Minor corrections

Line 239: change “suspense” to “suspensa”.

Line 247: either omit “there are” or add “that” in front of “could”.

Reviewer 4 Report

The submitted manuscript by Ying Yan and co-authors reported the genetic sexing system improvement of Lucilia cuprina by using a moderate driver and effector. Multiple TESS strains were obtained using different driver and effector. Fertility and lethality tests were performed to assess the performance of these TESS strains. The results showed that the “moderate” driver and effector lines could be an effective TESS strain that does not require tetracycline in the adult diet. The research is interesting and could be published in Insects.

The following points are listed for consideration:

  1. The title

The title is a little bit too long, if authors agree, it may be shortened as ‘Using moderate driver and effector strategy to improve the genetic sexing system of Lucilia cuprina’. However, this is just a suggestion. Authors could make decision.

  1. The key words

The current key words might be replaced by: Lucilia cuprina, sterile insect technique, genetic sexing, moderate driver/effector.

  1. strain stability

Please explain the stability of the TESS strain, DR2#7; EF3A.

  1. Spell check

In line 266, will it be correct to replace ‘DR2#8; EF#13’ with ‘DR2#8; EF-13’